# Rotating Single-Antenna Spoofing Signal Detection Method Based on IPNN

**DOI:** 10.3390/s22197141

**Published:** 2022-09-21

**Authors:** Haowei Chang, Chunlei Pang, Liang Zhang, Zehui Guo

**Affiliations:** 1Information and Navigation College, Air Force Engineering University, Xi’an 710077, China; 2Unit 94916 of PLA, Nanjing 211500, China

**Keywords:** forward spoofing, rotating single antenna differential detection Model, IPNN, smoothing factor

## Abstract

The traditional carrier-phase differential detection technology mainly relies on the spatial processing method, which uses antenna arrays or moving antennas to detect spoofing signals, but it cannot be applied to static single-antenna receivers. Aiming at this problem, this paper proposes a rotating single-antenna spoofing signal detection method based on the improved probabilistic neural network (IPNN). When the receiver antenna rotates at a constant speed, the carrier-phase double difference of the real signal will change with the incident angle of the satellite. According to this feature, the classification and detection of spoofing signals can be realized. Firstly, the rotating single-antenna receiver collects carrier-phase values and performs double-difference processing. Then, we construct an IPNN model, whose smoothing factor can be adaptively adjusted according to the type of failure mode. Finally, we use the IPNN model to realize the classification and processing of the carrier-phase double-difference observations and obtain the deception detection results. In addition, in order to reflect that the method has a certain practical value, we simulate the spoofing scenario of satellite signals and effectively identify abnormal satellite signals according to the detection results of the inter-satellite differential combination. Actual experiments indicate that the detection accuracy of the proposed method for spoofing signals reaches 98.84%, which is significantly better than the classical probabilistic neural network (PNN) and back-propagation neural network (BPNN), and the method can be implemented in fixed base station receivers for the real-time detection of forwarding spoofing.

## 1. Introduction

In recent years, with the development of Global Navigation Satellite System (GNSS) construction and service capabilities, related products have gradually penetrated into all aspects of social production and military development. It plays an important role in civil fields such as transportation, marine fishery, and electric power communication, and military fields such as military aviation, precision guidance, target reconnaissance, and security dispatch. However, due to the vulnerability of the GNSS signal itself, it is extremely vulnerable to artificial deception interference during the receiving process [1]. Common spoofing methods include production spoofing and forwarding spoofing [2]. Among them, the forwarding spoofing interference has strong concealment and is easy to implement. The principle is to first receive the real satellite signal, use the transponder to perform time delay and power amplification processing on the signal, and then radiate the signal through the transmitting antenna. Since the power of the forwarded spoofing signal is higher than the real signal, the real signal will be suppressed, so that the target receiver will use the forwarded signal as the real satellite signal to capture and track. At this time, the position coordinates obtained by the receiver are located near the transponder, which is related to the transponder delay. The existence of forwarding spoofing interference seriously threatens the security of navigation and positioning. Therefore, it is necessary to study effective spoofing detection methods according to the characteristics of forwarding spoofing interference [3,4,5,6,7].

At present, the commonly used spoofing detection methods are mainly based on the antenna array technology [8]. The principle is to use the difference in the spatial distribution of the real signal and the spoofed signal to realize the angle of arrival detection of the received signal. Under the condition of known receiver antenna array baseline and attitude, Ref. [9] proposes a method for spoofing detection using dual-antenna carrier-phase difference. Additionally, Ref. [10] proposes a construction method of phase double-difference spoofing jamming detection and verifies the minimum number of antennas to achieve double-difference detection. Based on the carrier-phase double-difference detection technology, Ref. [11] uses the azimuth change of the binary antenna array to make multiple judgments, which reduces the false alarm probability of detection. However, this kind of technology has high hardware complexity, long detection time, and the usage scenarios will also be limited to a certain extent [12]. Therefore, Nielsen and Broumandan proposed a method for spoofing detection using single-antenna motion [13,14], which is mainly based on the characteristic that the spatial correlation of spoofing signals is significantly stronger than that of real signals [15], and outputs the observation values of the moving antenna at different epochs to detect spoofing signals. On this basis, Ref. [16] realized the detection of spoofing signals by using the differential pseudorange observation method. The authors of [17] used the corrected pseudorange observations, carrier coordinates, and satellite coordinate information to calculate the pseudorange double difference and the error value of the double difference of the distance between the satellite and the Earth realizes the effective identification of abnormal satellites. However, this method needs to capture and track all signals at the same time, and combine the output results for detection, which greatly increases the complexity of the receiver algorithm. Moreover, since the antenna movement requires greater randomness, it cannot be directly applied to a base station receiver with a fixed and precise installation position. As commented in [18], when the receiver antenna moves, the receiver position changes, resulting in a change in the distance from the source of the signal, while the angle of arrival information of the incident signal is the main factor causing this distance change. Hence, by limiting the motion state of the receiver, a distance measurement value including the known motion and the angle of arrival of the unknown signal is obtained, and the measurement value is used to detect the angle of arrival of the spoofing jamming signal. Based on this, the paper [19] proposed to use the measured value of the output carrier phase of the rotating antenna for spoofing jamming detection and deduced a GLRT detection method based on the uncorrelated difference sequence. The paper [20] proposed a spoofing countermeasure based on the power measurements of a single rotating antenna according to the correlation coefficient of power measurement values. The above-mentioned method based on a rotating single antenna is mainly used to detect the existence of spoofing interference, and no in-depth research has been made on which satellite signal is in an abnormal state.

The Probabilistic Neural Network (PNN) was proposed by Specht according to Bayesian classification rules and Parzen probability density function. It has the advantages of simple structure, high training efficiency, and strong nonlinear identification ability, and has been widely used in fault diagnosis and classification detection [21]. Ref. [22] establishes a probabilistic neural network fault diagnosis model with clustered data as input, which solves the problem that the electrical characteristics of the array are difficult to accurately express under different fault conditions. Actual operational data are used to successfully verify the effectiveness and feasibility of the proposed method. Ref. [23] uses the PNN network to detect gas turbine sensor failure. The results show that the PNN model is able to detect sensor failures even in the presence of failed engine components as well as engine aging. However, the value of the smoothing factor of the traditional PNN model is fixed, ignoring its influence on different sample labels, resulting in a reduction in fault resolution, which in turn affects the final detection result. In order to solve this problem, [24] integrates the adaptive strategy into the PNN model and proposes a self-adaptive probabilistic neural network (SaPNN). In SaPNN, the best Spread can be self-adaptively selected all the time; thus, SaPNN can always get the best prediction accuracy. Ref. [25] further proposed an improved probabilistic neural network (IPNN) for solving the problem of low rate, which often exists in PNN, because PNN uses the same smooth factor during diagnosis procession. In IPNN, the smooth factor changes adaptively according to different categories of modes, so hidden neurons have high adaptabilities for function approximation, which better expresses the correlations between feature vectors and its pattern, better reflects the actual function of input feature vectors with final correct classification results, and the IPNN is applied to the fault diagnosis of rolling bearings.

This paper mainly focuses on the rotating single-antenna spoofing signal detection method based on IPNN. Firstly, an analysis of rotating single-antenna carrier-phase differential spoofing detection technology is performed, to obtain the relationship between the real signal carrier-phase double difference and the incident angle. Then, we introduce an adaptive algorithm to improve the traditional PNN model, and the value of the smoothing factor is adaptively adjusted according to different failure modes. On this basis, we input the double-difference value of the carrier phase obtained under the actual experimental conditions and perform spoofing detection on the observed value. Finally, through the comparison with the classic PNN model and BPNN model, the effectiveness of the algorithm is verified, and the effective identification of abnormal satellite signals is realized according to the results of classification detection.

## 2. Rotating Single-Antenna Carrier-Phase Differential Spoofing Detection Model

In this section, we describe the framework of the rotating single-antenna carrier-phase differential detection model and the principle of spoofing detection. 

### 2.1. Model Description

In this paper, we propose a single rotating antenna carrier-phase differential model that can detect spoofing signals. The model adds a rotating control structure on the basis of the existing receiver, then outputs the carrier-phase value when the single antenna is in a rotating motion state, so as to achieve the angle of arrival for spoofing. The detection model does not need to modify the signal processing algorithm of the receiver and can be directly used in the design of the spoofing detection scheme. Ref. [20] proposes a rotating single-antenna detection model, as shown in the Figure 1.

β in Figure 1 means the angle between the antenna stand and the horizontal plane. Based on this, this paper further studies a rotating single-antenna detection model based on carrier-phase difference. The pitch and azimuth angles of the satellite incident signal are used to represent the carrier-phase value. Its model is shown in Figure 2.

The phase center of the receiver antenna makes a uniform circular motion around the circle center O with an angular velocity ω (the position of the circle center is fixed and has been accurately calibrated in the ECEF coordinate system in advance), and the rotation radius is r. At this time, the center of the circle is used as the coordinate origin to establish a rectangular coordinate system, and the direction of the OX axis is from the origin O to the antenna phase center at time 0.

During the rotation process, the carrier-phase observations of the satellite ϕ(n) under N epochs are:(1)ϕ(n)=ρ(n)+ε(n)…n=1,2,3,⋯N−1
where ρ(n) represents the satellite carrier-phase true value and ε(n) represents the Gaussian white noise. It can be seen from paper [18] that ρ(n) can be further expressed as:(2)$$\rho (n)=\rho_{T}(n)+r\cos\theta \cos(\omega n+\psi )$$
where $\rho_{T}(n)$ represents the observed true value of the carrier phase at the origin O. θ and ψ are the pitch angle and azimuth angle of the satellite incident signal relative to the rotation plane xOy. Since the center O is stationary, in a short observation interval, the change of $\rho_{T}(n)$ is not affected by the spoofing jamming signal, and it can be considered that it is completely generated by the movement of the satellite.

Substitute (2) back into (1),
(3)$$\phi (n)=\rho_{T}(n)+r\cos\theta \cos(\omega n+\psi )+\varepsilon (n)$$

From the analysis of (3), it can be obtained that after the accurate calibration of the center position O, the carrier-phase measurement value sequence contains the information of the pitch angle θ and azimuth angle ψ of the incident signal.

### 2.2. Detection Principle

Assuming the carrier-phase observation value $\Phi ={\left[\begin{array}{ccc} \begin{array}{ccc} \phi (0) & \phi (1) & \phi (2) \end{array} & \cdots & \phi (N-1) \end{array}\right]}^{T}$ under N epochs, the Φ sequence not only contains the information of θ and ψ under each epoch but also has non-deterministic values with integer ambiguity, which makes it impossible to obtain angle information directly from the Φ sequence for angle of arrival detection. Therefore, in order to eliminate the influence of the whole-week ambiguity on deception detection, a forward-backward differential operation is performed on Φ:(4)$$\begin{array}{l} d\phi (k)=\phi (k+1)-\phi (k-1)\\ =\{[\rho_{T}(k+1)+r\cos\theta \cos[\omega (k+1)+\psi ]+\varepsilon (k+1)]-[\rho_{T}(k-1)+r\cos\theta \cos[\omega (k-1)+\psi ]+\varepsilon (k-1)]\}\\ =d\rho_{T}(k)-2r\sin\omega \cos\theta \sin[\omega k+\psi ]+\varepsilon_{n}(k)\\ k=1,2,3,\cdots ,N-2 \end{array}$$
where $\varepsilon_{n}(k)$ represents the new noise term obtained after the noise term is differentiated.

When spoofing jamming is performed on a receiver that has accurately calibrated the location of the fixed site, the ${d\rho }_{T}\left(k\right)$ of the spoofed signal and the real signal should be the same as the carrier-phase change caused by the satellite motion [26]. Hence, using the prior ephemeris data to calculate the satellite orbital motion, the influence of the ${d\rho }_{T}\left(k\right)$ can be eliminated, and the difference sequence $d\phi_{n}\left(k\right)$ containing only the carrier-phase observations and the noise term can be obtained:(5)$$\begin{array}{cc} d\phi_{n}(k)=-2r\sin\omega \cos\theta \sin[\omega k+\psi ]+\varepsilon_{n}(k) & k=1,2,3,\cdots ,N-2 \end{array}$$

Expressed in vector form:(6)$$d\Phi_{n}=H\Lambda +\varepsilon_{n}$$

Each vector in (6) can be expressed as:(7)$$H=-2r\sin\omega \left[\begin{array}{c} \begin{array}{cc} \;\sin(\omega ) & \cos(\omega ) \end{array}\\ \begin{array}{cc} \;\sin(2\omega ) & \cos(2\omega ) \end{array}\\ \begin{array}{cc} \;\sin(3\omega ) & \cos(3\omega ) \end{array}\\ \vdots \\ \begin{array}{cc} \sin\left[(N-2)\omega \right] & \cos\left[(N-2)\omega \right] \end{array} \end{array}\right]$$
(8)$$\Lambda =\left[\begin{array}{c} \cos\theta \cos\psi \\ \cos\theta \sin\psi \end{array}\right]$$
(9)$$\varepsilon_{n}={\left[\begin{array}{ccc} \begin{array}{cc} \begin{array}{cc} \varepsilon_{n}(1) & \varepsilon_{n}(2) \end{array} & \varepsilon_{n}(3) \end{array} & \cdots & \varepsilon_{n}(N-2) \end{array}\right]}^{T}$$

In the real state, the pitch angle θ and the azimuth angle ψ of different satellite signals are different when they reach the rotating plane; while, for the multi-channel spoofing signals transmitted by a single antenna, the arrival angles of the signals are exactly the same. According to this feature, the carrier-phase sequences of the two signals i and j are differentiated, and we then obtain the double-difference observation equation:(10)$$\begin{array}{l} \Delta d\Phi_{n}{}^{ij}=d\Phi_{n}{}^{i}-d\Phi_{n}{}^{j}=H(\Lambda^{i}-\Lambda^{j})+(\varepsilon_{n}^{i}-\varepsilon_{n}^{j})=HR^{ij}+\Delta d\varepsilon_{n}^{ij} \end{array}$$

When the two satellite signals i and j are real signals, with the constant changes of θ and ψ of the two signals, $R^{ij}$ also changes accordingly. When the satellite signals i and j are spoofed signals, $R^{ij}$ is equal to 0, and the observation equation only contains the noise term. According to this feature, the signal types can be distinguished, and the detection of deception signals can be realized.

## 3. Rotating Single-Antenna Spoofing Signal Detection Method Based on IPNN

In this section, based on the discussion of the classical PNN network structure and the principle of classification detection, in view of the problem that the fixed smoothing factor in the detection process leads to the reduction of the recognition rate, an adaptive function is introduced to adjust the smoothing factor, so that it changes dynamically according to the different types of failure modes. Finally, the steps of using the improved probabilistic neural network for deception signal detection are given. 

### 3.1. PNN Network Model

PNN is a feedforward neural network algorithm developed by combining the Bayesian criterion and Parzen probability density function [27]. As a branch of a radial basis network, PNN network has strong nonlinear ability recognition, which is widely used in solving pattern classification and fault diagnosis problems and has advantages in the detection of deception signals [28].

The PNN model is based on Bayes decision theory to realize the classification and judgment of fault signals. Assuming a set of *n* dimensional vectors $X=[x_{1},x_{2},\;\cdots \;x_{n}]$ under different categories $\varphi_{1},\varphi_{2},\cdots ,\varphi_{p}$, according to the optimal classification principle of Bayesian minimum “expected risk”, the conditions for judging $X\in \varphi_{p}$ are:(11)$$h_{p}l_{p}f_{p}(X)>h_{q}l_{q}f_{q}(X)\;(p\neq q)$$
where $h_{p}$ and $h_{q}$ represent the prior probabilities of occurrence of $\varphi_{p}$ and $\varphi_{q}$, $l_{p}$ and $l_{q}$ represent the loss function caused by misjudgment of $\varphi_{p}$ and $\varphi_{q}$, $f_{p}(X)$ and $f_{q}(X)$ represent the probability density function of $\varphi_{q}$ and $\varphi_{q}$.

In the detection process, the prior probability h and the loss function l should be the same for both the real signal and the deceptive signal. Therefore, only the probability density function f(X) needs to be determined to achieve the purpose of classification and detection.

PNN adopts the nonparametric estimation method of the Parzen window to determine the probability density function:(12)$$f_{p}(X)=\frac{1}{k}\frac{1}{{\left(2\pi \right)}^{n/2}\sigma^{n}}\sum_{i=1}^{k}\exp\left[-\frac{{\left(X-X_{pi}\right)}^{T}(X-X_{pi})}{2\sigma^{2}}\right]$$
where k represents the total number of training samples under $\varphi_{p}$, $X_{pi}$ represents the *i*-th training sample of $\varphi_{p}$, and σ is the smoothing factor.

As can be seen in Figure 3, the PNN model consists of four parts: input layer, pattern layer, addition layer, and output layer. The input layer is used to receive input samples, and, after normalization, the sample data is passed to the network, and the number of neurons is the same as the dimension of the sample feature vector. The pattern layer first analyzes the matching relationship between the input sample feature vector and the different sample labels in the training set and then sends it to the node activation function in the form of Euclidean distance to obtain the output of each neuron. The node activation function of this layer $h_{pi}(X)$ is:
(13)$$h_{pi}(X)=\exp\left[\frac{X^{T}X_{pi}-1}{\sigma^{2}}\right]$$

The addition layer weighs and sums the outputs of neurons under the same category in the pattern layer, then calculates $f_{p}(X)$ to obtain the probability density function corresponding to the sample data and the label. The output layer is composed of a threshold comparator. According to the probability density values output by different types of neurons in the addition layer, the signal category to which the observed sample belongs is judged, and the label type corresponding to the neuron with the highest probability density is selected as the signal type of the sample, so as to obtain deception test results.

### 3.2. IPNN Basic Principles

The traditional PNN model often uses the same smoothing factor for different sample labels, ignoring the degree of its influence on the samples, resulting in a lower fault recognition rate. In the actual detection process, if the smoothing factor is too small, the fault detection cannot be performed effectively; if the smoothing factor is too large, the detection accuracy will decrease [25]. In response to this problem, an improved probabilistic neural network (IPNN) is proposed to improve the classical PNN, that is, an adaptive iterative algorithm is introduced to optimize the smoothing factor, so that it can adjust the value range adaptively according to the signal type, thereby improving the spoofing detection performance. Its expression is:(14)$$\sigma_{i}(k)=\sigma_{i}(k-1)+\eta \sigma (k)+\lambda [\sigma_{i}(k-1)-\sigma_{i}(k-2)]$$
where η denotes learning step size, 0<η<0.1, λ is the momentum factor, and 0<λ<0.001. Through the adaptive change of the smoothing factor, the probability density function has better ductility and variability, overcomes the independent and identical distribution assumption of the classical PNN for the input data, enhances the correlation between the sample feature vector and the fault type, and the efficiency and accuracy of fault identification of the model are improved.

### 3.3. Spoofing Jamming Detection Process

Using the IPNN model constructed to detect spoofing signals is shown in Figure 4. There are six procedures in the whole process:

(1)Using the rotating single-antenna model, the carrier-phase observations under the real signal and the spoofed signal are collected respectively, and the double-difference processing is performed on them:(2)The double-difference observations under the real signal are mainly affected by the values of θ and ψ, while the double-difference observations under the deceptive signal are only related to the noise term. According to this feature, the sample label is set. The real signal label is 1, and the deception signal label is 2;(3)Initialize the network weights, input the training samples, that is, the carrier-phase double-difference observations, and send the sample feature vectors to the pattern layer after normalization;(4)Analyze and calculate the errors of the samples under different deception models, obtain the smoothing factors under different sample labels by (14), and then establish the corresponding Gaussian kernel function as the node activation function of the network;(5)Input the test sample, use the trained IPNN network for deception detection, and get the deception detection result;(6)Simulate the real situation where the satellite is deceived and interfered, and through the IPNN classification and detection results, the abnormal satellite signals can be eliminated.

## 4. Experimental Analysis

In this section, the effectiveness of the proposed spoofing detection method is verified by conducting real GNSS spoofing attack experiments on real-time receivers. The experimental scene is shown in Figure 5. The forwarding spoofing device consists of a GNSS signal transponder, a receiving antenna, and a spoofing signal transmitting antenna. The spoofing detection platform consists of a NovAtel702 receiving antenna with a rotating control platform, a receiver, and a data processing platform. We set the angular velocity ω=12∘s of the rotary table, and the sampling frequency is 1 Hz.

### 4.1. Data Collection and Processing

The experiment is divided into two completions. In order to ensure the validity of the experimental data collection, each group of experiments should follow the same ephemeris environment, and the transponder parameters should also be consistent in the deception experiments. The first experiment consists of two groups of experiments, mainly to complete the collection of training data, which are:(a)Turn on the receiving device; all received satellite signals are real signals;(b)The receiving device is placed in the house, and the signal is received by the receiving antenna placed outdoors. After being processed by the signal transponder, it is transmitted to the receiver by the transmitting antenna indoors for collection, thus, ensuring that the data collected by the receiver are all spoofing signals.

The second experiment was to collect test data. In order to simulate the forwarding spoofing interference in real situations, the scheme is designed as follows:
(a)The receiving device is working normally, the GNSS signal transponder is not working, and the receiver receives the real signal at this time;(b)Turn on the signal transponder and keep the receiving device working normally. At this time, the collected data are changed from real signal to spoofing signal.

Each set of experiments took 30 min. The two experiments collected 3457 and 3362 effective carrier-phase observation data points of each satellite, respectively. The number of satellites and their distribution information visible during the experiment are shown in Figure 6.

We performed double-difference processing on the above eight visible satellites to obtain 96,740 training data and 94,080 test data points. We define every 100 pieces of data as a group of samples, then, we have 967 groups of training samples and 940 groups of test samples, in which the sample label category and the capacity are shown in Table 1.

### 4.2. Analysis of Experimental Results

Under the adjustment of the adaptive function, the smoothing factor changes its value according to the spoofing mode, and then generates activation functions under different models, which avoids the degradation of detection performance caused by the reduction of the resolution of the classification model.

A total of 300 sets of data are randomly selected from the training samples for neural network training, and 100 sets of data are randomly selected from the test samples for neural network testing. First, we determine the IPNN network structure. The number of neurons in the input layer is 3, which corresponds to the three-dimensional feature vector of the satellite signal pitch angle θ, azimuth angle ψ, and noise term in (10); the number of neurons in the pattern layer is 300, which corresponds to 300 groups of training samples; the addition layer sets the number of neurons to 2 according to the two failure modes of real and deception; the output layer only needs 1 neuron to output 1 or 2 to represent the final detection result, so the IPNN network structure is 3-300-2-1. Then, ten epochs of data are continuously selected from each group of samples and input into the detection model, and the corresponding smoothing factor σ under each deception model is obtained by analyzing and calculating the error of the eigenvectors under different failure modes. According to the obtained smoothing factor, the corresponding Gaussian kernel function is established and used as the node activation function of the network. The detection results and training errors of the training samples are shown in Figure 6.

Figure 7 shows that, after the adaptive adjustment of the IPNN network, the detection results of only 9 sets of data in the randomly selected 300 sets of sample data are inconsistent with the actual state, and the detection accuracy rate reaches 97%. Among them, when 254 groups of data containing spoofed signals were detected, 248 groups could accurately detect the difference between them and the real signal, and the detection success rate was 97.63%. Therefore, the IPNN network trained by the training samples has been able to effectively discriminate data types and complete the detection of spoofing signals. Then, the deception detection is performed on the test sample, and the detection result is shown in Figure 8:

By analyzing the test results of the above 100 sets of test samples, it can be concluded that under only ten epochs of observation data the detection success rate of the IPNN model for spoofing signals can reach 97.46%, and the detection success rate for real signals is 95.65%. Then, the signal accuracy in different modes is obtained as shown in Table 2.

In order to further verify that the IPNN model has a significant improvement in deception detection performance compared with the traditional fault detection model, 500 sets of data were randomly selected from the test sample, and the three models shown in Table 3 were used to select different epochs for multiple detections.

Among them, the number of input and output nodes of the BPNN model is consistent with that of the PNN and IPNN models, and the number of hidden layer nodes is seven according to Kolmogorov’s theorem; the smoothing factor σ in the PNN model takes 0.1 according to experience; when η takes 0.02 and λ takes 0.0004 in the IPNN model, the fault identification effect is the best. The final detection results and time are shown in Table 4.

We plotted the data as shown in Figure 9.

As the number of epochs increases, the accuracy rates of the three detection models gradually increase, reaching the maximum value when the number of sample epochs is 100, and the detection time of each model does not exceed 1 s, which can basically meet the requirements of real-time detection. When the training samples and test samples are the same, compared with the BPNN and PNN models, the IPNN model has the best detection effect and the shortest time. When the number of sample epochs reaches 100, the detection accuracy of the IPNN model reaches 98.84%, the PNN model is 94.95%, and the BPNN model is only 86.42%; the IPNN model has a faster convergence speed, and the training time is also significantly improved. The detection time is only 0.0963 s, which is 49.1% of PNN and 11.4% of BPNN.

## 5. Application Verification

On the basis of detecting spoofing interference, the IPNN model can further determine the signal type of the satellite through the classification results. As shown in Figure 10, the navigation signal source simulates the signals of five visible satellites, and the spoofing jamming source deceives and jams several of the satellites according to the deception strategy.

The signal types of the visible satellites are set as shown in Table 3.

In Table 5, R represents the real signal and S represents the deceitful signal. After the carrier-phase observation data collected by the receiver were subjected to double-difference processing, they were input into the IPNN network for detection. The detection results are shown in Table 6.

The five satellite signals were combined in pairs to obtain ten sets of inter-satellite differential combinations, then, we used the IPNN network for detection. When the detection result is 1, it means that the two satellite signals are real signals, and when the detection result is 2, it means that there is a spoofing signal in the two satellite signals. In group 1, it can be found that the No. 2, No. 3, No. 8, and No. 18 satellites received real signals from S02 and S03, S02 and S08, and S02 and S18. Then, we analyzed other combinations and the No. 5 satellite was found to be interfered by the spoofing signal. In group 2, the No. 3, No. 5, and No. 18 satellites received real signals from S03 and S05 and S03 and S18, and it can be seen that the satellite signals of No. 2 and No. 8 were spoofing signals from the other combinations. In group 3, the No. 2, No. 5, and No. 8 satellites received real signals from S02 and S05 and S02 and S08, and it can be found that the signal data of the No. 3 and No. 18 satellites were abnormal from S02 and S03 and S02 and S18. According to the classification and detection results, it is possible to effectively discriminate the types of satellite signals and ensure the authenticity and reliability of the observation data.

The above experimental results show that the method proposed in this paper can effectively detect the spoofing signal emitted by a single spoofing source, and can effectively screen out abnormal satellites, which can be used for anti-spoofing and jamming modification of various fixed site receivers. However, with the increasing development of deception technology, advanced deception jamming technology emerges as the times require. This technology adopts the mode of cooperative deception of multiple deception signal sources, which highly restores the distribution of real satellite signals in space, and is unable to achieve spoofing detection technology based on the signal angle of arrival. Therefore, on this basis, research on anti-jamming technology for multi-spoofing source deception should be carried out.

## 6. Conclusions

In order to solve the GNSS forwarding spoofing jamming problem, this paper proposes a rotating single-antenna spoofing jamming detection method based on IPNN. The following conclusions can be drawn:(1)When using the rotating single-antenna carrier-phase differential detection model to obtain double-difference observations, there is no need to rely on traditional antenna array technology, which reduces the cost of observation.(2)The IPNN model can accurately classify different differential signal types, and its detection accuracy and detection rate are significantly better than the BPNN model and the traditional PNN model.(3)Under the premise of detecting spoofing interference, the classification and detection results of the IPNN model can accurately determine the satellites containing spoofing signals, which is suitable for forwarding spoofing interference detection of fixed-site receivers.

## Figures and Tables

**Figure 1 sensors-22-07141-f001:**
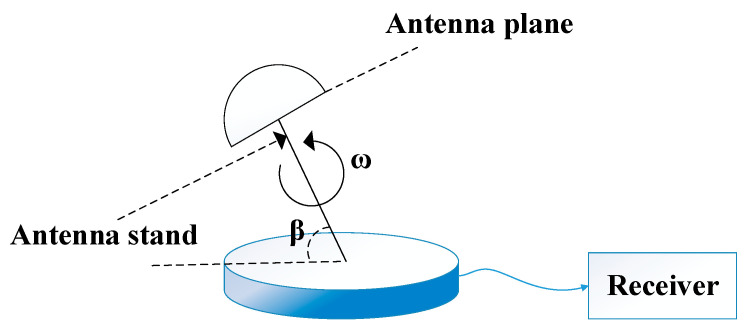
Rotating single-antenna detection model.

**Figure 2 sensors-22-07141-f002:**
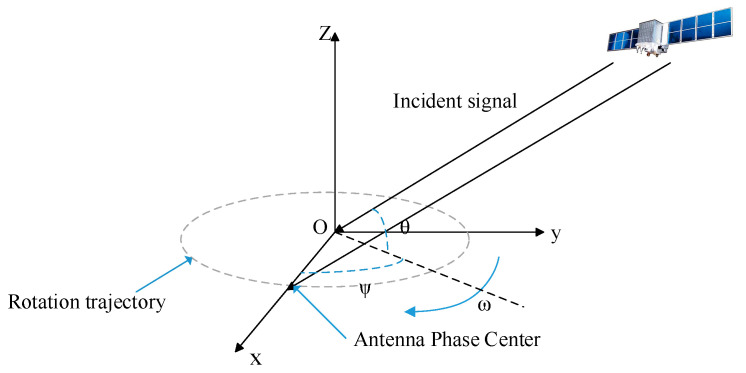
Rotating single-antenna carrier-phase differential detection model.

**Figure 3 sensors-22-07141-f003:**
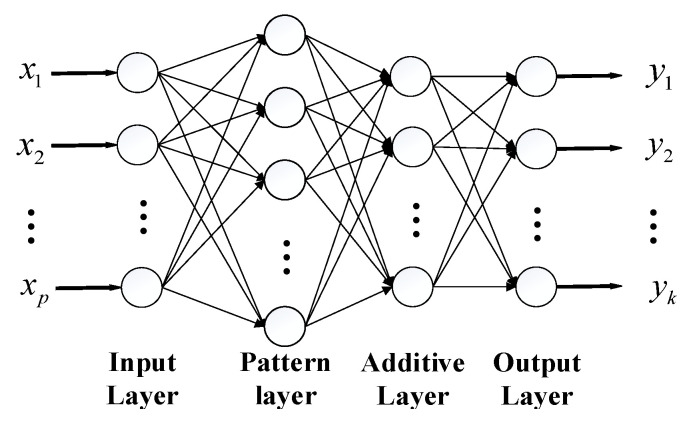
PNN model structure.

**Figure 4 sensors-22-07141-f004:**
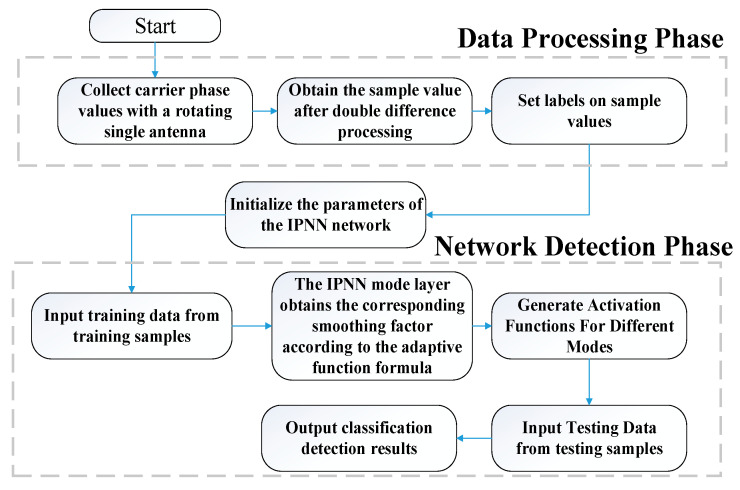
Spoofing Detection Process.

**Figure 5 sensors-22-07141-f005:**
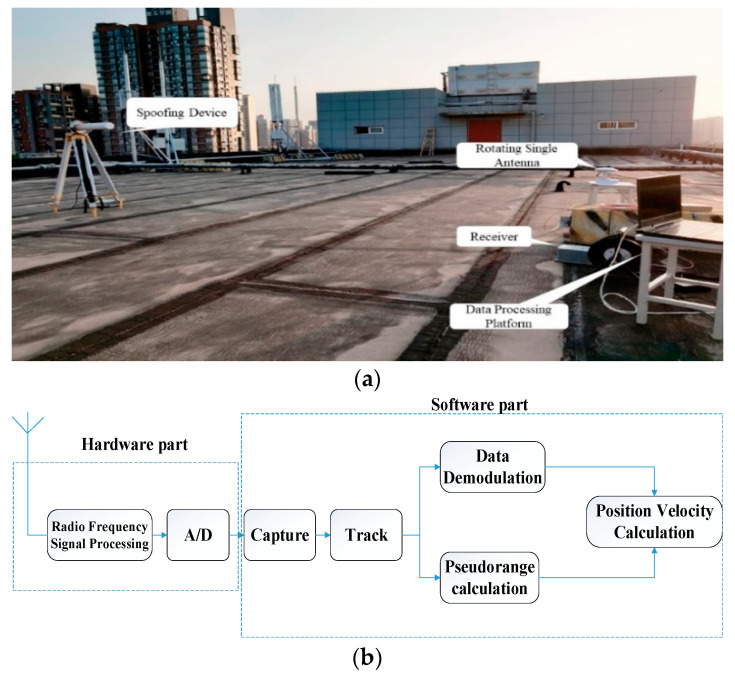
Experimental scene settings. (**a**) Rooftop experimental scene settings; (**b**) The working principle of the receivers.

**Figure 6 sensors-22-07141-f006:**
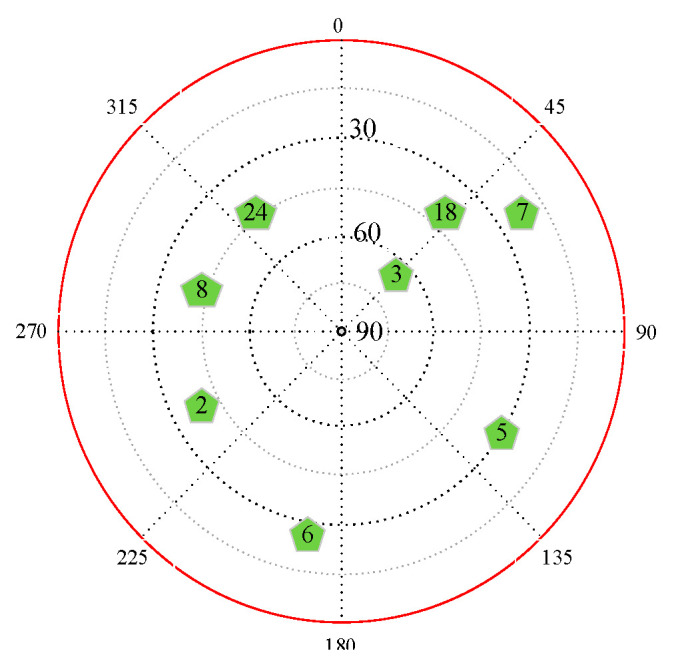
GNSS star map. The green pentagons represent the observed satellite signal, and the number above represents the satellite serial number.

**Figure 7 sensors-22-07141-f007:**
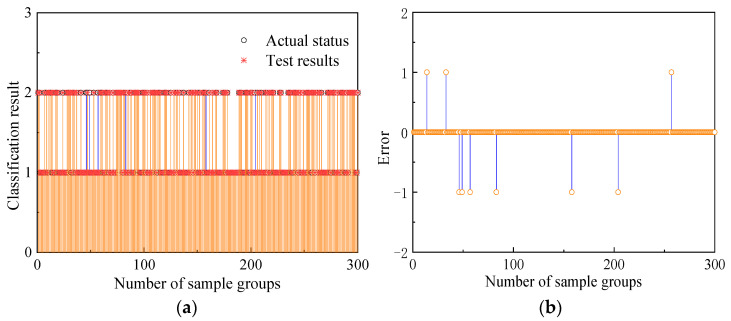
IPNN training effect diagram: (**a**) IPNN training data classification results; (**b**) IPNN training data detection error.

**Figure 8 sensors-22-07141-f008:**
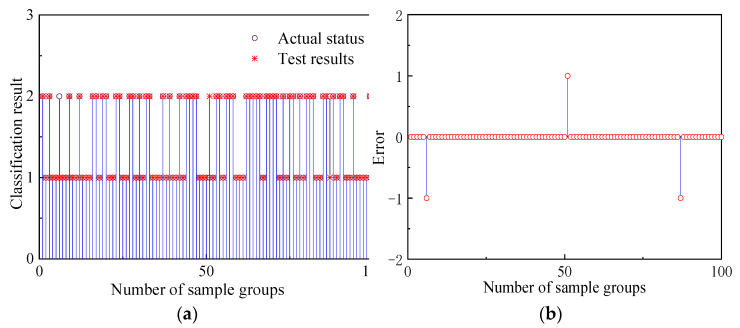
IPNN test effect diagram: (**a**) IPNN test data classification results; (**b**) IPNN test data detection error.

**Figure 9 sensors-22-07141-f009:**
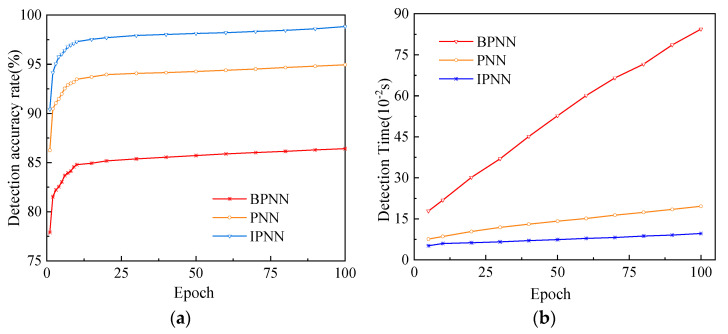
Detection performance change curve of three deception detection models: (**a**) Detection accuracy rate change curve of three deception detection models; (**b**) Detection time change curve of three deception detection models.

**Figure 10 sensors-22-07141-f010:**
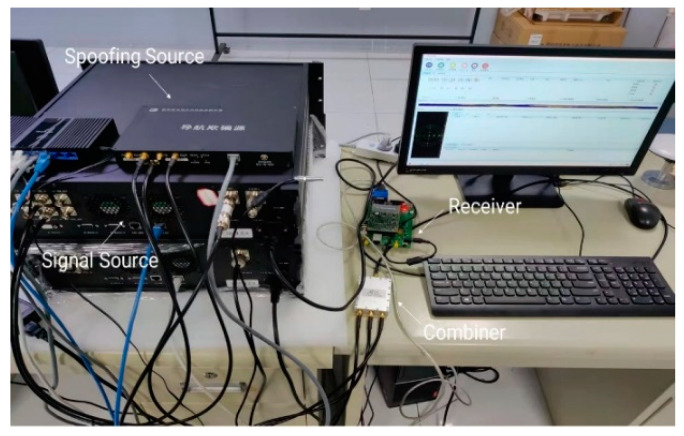
Simulation environment.

**Table 1 sensors-22-07141-t001:** Sample data settings.

Species	Authentic Signal	Spoofing Signal
Tag	1	2
The size of training sample	445	522
The size of testing sample	512	428

**Table 2 sensors-22-07141-t002:** The accuracy of signals under different models.

Signal Type	Detection Accuracy Rate(%)	
	Training Samples	Testing Samples
Real signal	93.48	95.65
Spoofing signal	97.63	97.46

**Table 3 sensors-22-07141-t003:** Deception detection model.

Model	Parameter Setting
BPNN	Network structure: 3-7-1
PNN	Network structure: 3-300-2-1, σ=0.1
IPNN	Network structure: 3-300-2-1 η=0.02, λ=0.0004

**Table 4 sensors-22-07141-t004:** Final detection results and time.

		Detection Accuracy Rate (%)			Detection Time (10^−2^ s)		
		BPNN	PNN	IPNN	BPNN	PNN	IPNN
**Epoch**	5	83.67	91.98	96.03	17.8	7.59	5.14
	10	84.78	93.47	97.31	21.8	8.52	5.97
	20	85.17	93.95	97.71	30.1	10.35	6.24
	30	85.38	94.07	97.93	36.9	11.87	6.57
	40	85.54	94.15	98.03	45.1	13.05	7.03
	50	85.71	94.26	98.14	52.6	14.14	7.39
	60	85.88	94.39	98.22	60.1	15.12	7.84
	70	86.02	94.55	98.34	66.5	16.32	8.17
	80	86.14	94.67	98.45	71.5	17.38	8.69
	90	86.09	94.81	98.61	78.6	18.43	9.05
	100	86.42	94.99	98.84	84.4	19.58	9.63

**Table 5 sensors-22-07141-t005:** Satellite signal type.

Groups	Signal Settings				
	S02	S03	S05	S08	S18
G1	R	R	S	R	R
G2	S	R	R	S	R
G3	R	S	R	R	S

**Table 6 sensors-22-07141-t006:** IPNN network detection results.

Intersatellite Difference Combination	Detection Results		
	G1	G2	G3
S02 and S05	2	2	1
S02 and S03	1	2	2
S02 and S08	1	2	1
S02 and S18	1	2	2
S03 and S05	2	1	2
S03 and S08	1	2	2
S03 and S18	1	1	2
S05 and S08	2	2	1
S05 and S18	2	1	2
S08 and S18	1	2	2

## Data Availability

The data used to support the findings of this study are available from the corresponding author upon request.

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
