# Peer review of "Rotating Single-Antenna Spoofing Signal Detection Method Based on IPNN"

_sensors, 2022, doi:10.3390/s22197141_

Round 1
Reviewer 1 Report
sensors-1908644-peer-review-v1
Rotating single-antenna spoofing signal detection method based on IPNN
By Haowei Chang 1*, Chunlei Pang 1, Liang Zhang 1, ZeHui Guo
This paper proposed a rotating single-antenna spoofing signal detection method based on IPNN for static single-antenna receiver. The proposed method can adaptively adjust the smoothing factor regarding to different failure modes for a more accurate spoofing signals’ detection. The effectiveness of the method is verified by comparisons of classic PNN model and BPNN model. The paper is well prepared. I have the following comments for your consideration.
1. In the abstract, I suggest the authors to split the long description of proposed method into two more sentences, since the long sentence from Line 11 to Line 18 is tough to deliver the correct logic for readers. The same problem also happens in the introduction part from Line 64 to Line 67, which is repeated and unconcise.
2. In the introduction, I suggest the authors add more contents to review the applications of PNN and discuss their defects on paragraph Line 56 to Line 62. The benefits and applications of IPNN haven’t been discussed in the introduction part. I suggest authors to provide more support from existing research to make introduction comprehensively.
3. In part 2 and 3, please make all the equations in line with the words.
4. There is no reference in section 2.1 model description part. Please add reference before the modeling.
5. In figure 5, there is no comprehensive explanation for the process flow. The term on the figure is expected to be well identified.
6. The conclusion part can also be further polished. There is no need to make to many paragraphs here. The authors can just list their conclusion with 1)… 2)… 3)…
Author Response
Dear doctor,
First of all, thank you for your comments. I have made the following changes in response to your comments.Please see the attachment for details.

Reviewer 2 Report
The paper titled "Rotating single antenna spoofing signal detection method based on IPNN" proposes an adaptive algorithm in order to enhance the PNN model and to get the accurate detection of spoofing single by adjusting the spoofing factor in accordance with the failure modes. The paper represents useful and effective results by identifying the abnormal signal from satellites that have been realized in accordance with the classification detection results. This paper possesses great importance for the studies related to detecting spoofing signals, however. There are some comments/suggestions that should be adopted to add value to the paper.
- Forward spoofing is discussed briefly. Details like, what is forward spoofing? And how does it work? Should be added in parallel with its effects for a clearer understanding for the reader.
- The paper includes 2 to 3 literature reviews of the related study, I would suggest adding a few more papers (3-4) literature to make the readers distinguish and analyze the impact of this study.
- I would suggest adding the block diagram for a pictorial illustration of the working principle done in this study. (i.e. the representation of the working mechanism of the receivers, antenna etc. as shown in figure 3)
- A separate paragraph/table should be added to compare the actual detection process and the process proposed in the study.
- A tabular representation should be added to show the accuracy of signals under different models.
- Future challenges and advancements in the proposed study should be discussed.
- I would suggest discussing all the results presented in table 4.
Additional Suggestions
8. The images are not aligned with the text and must have the same size and formatting
9. There are some grammatical errors which are needed to be corrected.
Author Response
Dear doctor,
First of all, thank you for your comments. I have made the following changes in response to your comments.Please see attachment for details.

Reviewer 3 Report
1. What are IPNN and BPNN? The authors must explain the meaning of abbreviations when mentioned for the first time.
2. The introduction must enhance the works in the literature explaining the gaps and shortcuts.
3. The introduction must contain some of the results of the work.
4. Mention the standard that the experiments follow.
5. What is the GNNS signal?
6. Can the authors review the format style of the journal?
Author Response

(The authors gave the same response as above.)

Round 2
Reviewer 2 Report
now it should be accepted.